# A Nanoporous Supramolecular Metal–Organic Framework Based on a Nucleotide: Interplay of the π···π Interactions Directing Assembly and Geometric Matching of Aromatic Tails

**DOI:** 10.3390/molecules26154594

**Published:** 2021-07-29

**Authors:** Rosaria Bruno, Teresa F. Mastropietro, Giovanni De Munno, Donatella Armentano

**Affiliations:** Dipartimento di Chimica e Tecnologie Chimiche (CTC), Università della Calabria, Rende, 87036 Cosenza, Italy; rosaria.bruno@unical.it (R.B.); teresafina.mastropietro@unical.it (T.F.M.); giovanni.demunno@unical.it (G.D.M.)

**Keywords:** supramolecular chemistry, metal-organic frameworks, nucleotide-based materials, chirality

## Abstract

Self-assembly is the most powerful force for creating ordered supramolecular architectures from simple components under mild conditions. π···π stacking interactions have been widely explored in modern supramolecular chemistry as an attractive reversible noncovalent tool for the nondestructive fabrication of materials for different applications. Here, we report on the self-assembly of cytidine 5’-monophosphate (CMP) nucleotide and copper metal ions for the preparation of a rare nanoporous supramolecular metal-organic framework in water. π···π stacking interactions involving the aromatic groups of the ancillary 2,2’-bipyridine (bipy) ligands drive the self-assemblies of hexameric pseudo-amphiphilic [Cu6(bipy)6(CMP)2(µ-O)Br4]^2+^ units. Owing to the supramolecular geometric matching between the aromatic tails, a nanoporous crystalline phase with hydrophobic and hydrophilic chiral pores of 1.2 and 0.8 nanometers, respectively, was successfully synthesized. The encoded chiral information, contained on the enantiopure building blocks, is transferred to the final supramolecular structure, assembled in the very unusual topology **8****T6**. These kinds of materials, owing to chiral channels with chiral active sites from ribose moieties, where the enantioselective recognition can occur, are, in principle, good candidates to carry out efficient separation of enantiomers, better than traditional inorganic and organic porous materials.

## 1. Introduction

The realm of chemistry beyond molecules was unveiled by Pedersen, Lehn, and Cram, who were awarded with the Nobel prize in 1987 for their pioneering works [1,2,3]. Since then, the chemistry of supramolecular assemblies, constituted by discrete components held together by means of intermolecular interactions, has greatly expanded, covering a different area of applications and definitely transferring the fundamental paradigms of molecular biology to materials science [4,5,6,7,8].

Hydrogen bonds, π-π stacking, hydrophobic segregation, electrostatic and van der Waals forces have been widely explored for applications in modern supramolecular chemistry. Due to the unique advantages, they are of the non-destructive, reversible, and modular fabrication of hierarchically ordered structures [9,10,11,12,13]. Despite being relatively weak with respect to covalent bonds [14,15], intermolecular interactions can cooperatively control and stabilize the assembly of discrete supramolecular systems, as well as extended three-dimensional networks [16,17]. In particular, the self-assembly of discrete units offers a powerful tool for creating supramolecular extended architectures from simple components under mild conditions and provides a higher degree of tunability, enabling the fine tailoring of their structures as well as their chemical-physical properties [18]. 

Hydrogen bonds are the leading intermolecular interactions exploited by supramolecular engineers to stabilize extended supramolecular networks. An outstanding example is constituted by a recently emerged class of crystalline porous materials namely hydrogen-bonded organic frameworks (HOFs) and hydrogen-bonded metal-organic frameworks (HMOFs), generally included in HOFs [19,20,21,22]. In contrast to classic metal-organic frameworks (MOFs) [23,24,25,26,27,28] or covalent organic frameworks (COFs) [29,30], HOFs and HMOFs are conventional supramolecular compounds mainly stabilized through hydrogen bond interactions. With respect to the more rigid systems of MOFs and COFs, HOFs and HMOFs offer some remarkable opportunities such as greener synthesis conditions, easier solution processability, practical healing, and regeneration which make them very attractive candidates for the construction of functional materials [31,32,33]. Nevertheless, even if H-bonds dominate their assembly, other forces coexist and sometimes direct the supramolecular construction, but their contribution has often been underestimated. Despite being more challenging, examples of porous supramolecular networks mainly sustained by π-π interactions have been increasingly reported in the literature, definitely demonstrating that they are non-negligible forces for stabilizing porous supramolecular frameworks, and often play a key role in determining their properties [34,35,36,37,38,39]. 

Among the plethora of available linkers, the employment of nucleic acids and their derivatives in nanotechnology is gaining increasing attention [40,41,42,43,44,45,46,47,48,49,50]. The presence of binding sites for metal ions and acceptor and donor functionalities for H-bonds make this class of molecules highly attractive in the field of supramolecular chemistry [42,49,51,52,53,54,55,56,57]. Obtained assemblies display controllable morphology and peculiar chemical–physical properties [58,59]. They often feature stimuli-responsive characteristics suitable for application in sensing, bio-imaging, drug delivery, and construction of logic gates [43]. Moreover, the improved compatibility and low toxicity of these biomaterials with respect to conventional ones make them a safer choice for biomedical applications [48,60]. The rational synthesis of these systems attracted high attention since the reciprocal interactions between nucleobases or with other small molecules and metals open to a wide range of synthetic strategies. The preparation of nucleotide–metal complexes with controllable final architecture has been a problem worthy of attention since time (see Appendix A) [61,62,63,64,65,66,67,68]. In parallel to this topic, the modulated synthesis is a very useful approach. Modulated synthesis refers to the regulation of the coordination equilibrium by the employment of modulators, which competitively with linkers coordinate the metals or might influence the electron doublet availability in linkers. As a result, the competitive reaction can reduce the rate of nucleation and slow down crystal growth to help produce highly crystalline products [69]. 

Within the framework of our works on the use of enantiopure nucleosides- [51,70,71,72] or nucleotides-based metal complexes [52,73] to generate supramolecular architectures of different dimensionality, we report here on the self-assembly of cytidine 5’-monophosphate (CMP) biomolecule and copper metal ions for the preparation of a unique supramolecular metal-organic framework with the formula [Cu_6_(bipy)_6_(CMP)_2_(µ-O)Br_4_]·Br_2_·46H_2_O (**CMP-MOF 1**). 

## 2. Results and Discussions

A pseudo-amphiphilic metal-based unit is generated from CMP nucleotide and copper metal ions by introducing 2,2’-bipyridine (bipy) as a modulator (Scheme 1). In **CMP-MOF 1**, the supramolecular organization involves mainly π···π stacking interactions capable of driving the self-assembly of hexameric [Cu_6_(bipy)_6_(CMP)_2_(µ-O)Br_4_]^2+^ units and transfer the chiral information from the building units to the resulting frame. Owing to the geometric matching between the supramolecular self-assembled aromatic tails, a nanoporous crystalline phase with hydrophobic and hydrophilic chiral pores was successfully synthesized (Figure 1, Figure 2, Figure 3 and Figure 4). Single-crystal X-ray Diffraction (SCXRD) unveiled the supramolecular architecture, featuring a very unusual topology 8T6 of the network, which shows chiral channels decorated with chiral active sites from the ribose moieties (Figure 1a and Figure 3). The here reported **CMP-MOF 1**, falls in the family of chiral porous materials, which have emerged as an ideal platform for application in heterogeneous enantioselective catalysis as well as for the selective recognition and separation of enantiomers from racemic mixtures since they can be opportunely engineered to provide the well-defined stereospecific environment required for the discrimination and separation of enantiomers. Furthermore, they can be readily recovered from the reaction medium and regenerated for reuse in successive cycles, thus enabling easy and low-cost processes. 

The **CMP-MOF 1** crystallizes in the *P*4, the space group of the tetragonal system, and its absolute configuration was reliably assigned (Flack parameter of 0.089(4), Appendix A). The structure consists of a pretty supramolecular assembly, made up of hexameric [Cu6(bipy)6(CMP)2(µ-O)Br4]^2+^ entities (Figure 1c), self-assembled by van der Waals interactions (Figure 1b). 

Pairs of μ_4_-phosphate groups connect four copper(II) ions, giving rise to butterfly-shaped tetranuclear cores of the type [Cu_4_(μ_4_-PO_4_)_2_(μ-O)], supported by a bridging oxo group (Figure 1c). Water molecules and oxo groups, the latter coordinating in a μ_2_ fashion, complete the square pyramidal geometry on metal ions. Oxygen atoms belonging to each phosphate connect the four metal ions through a double bridge, where one oxygen coordinates in a µ_2_-fashion (either Cu1 and Cu4 or Cu2, and Cu3 pairs, in Figure 1c), while the remaining two oxygen atoms individually bond a single metal atom. This phosphate coordination mode is usually observed in polynuclear compounds of cytidine monophosphate [52]. The terminal copper atoms, chelated by the cytosine base of the nucleotide molecules, constitute the terminal tails, acting as perfect connectives for intermolecular π···π stacking interactions, involving the ancillary bipyridine ligands, which drive the self-assembly (Figure 1b). Hence, the cytosine nucleobases of the two crystallographically distinct CMP ligands connect via N3-O2, the tetranuclear core grafting two additional copper metal ions, that complete their distorted octahedral geometry through the link of a bipy modulator and two bromides (Figure 1c).

**Figure 1 molecules-26-04594-f001:**
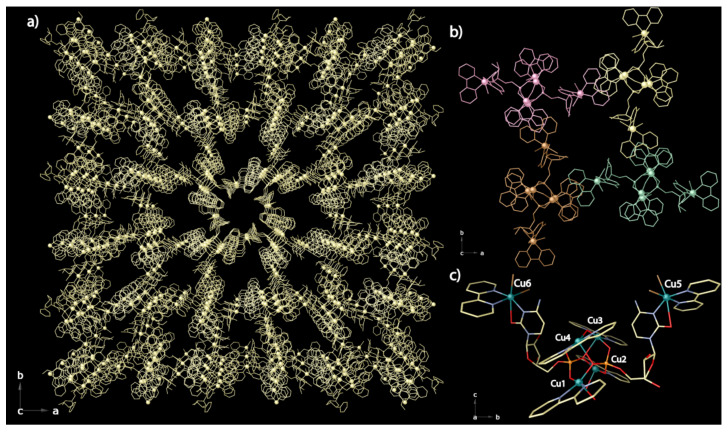
Crystal structure of **CMP MOF 1**: (**a**) perspective view of the 3D supramolecular porous network along the *c* crystallographic axis; (**b**) unit cell content representing four hexameric [Cu_6_(bipy)_6_(CMP)_2_(µ-O)Br_4_]^2+^ entities, packed via π···π interactions; (**c**) perspective view of [Cu_6_(bipy)_6_(CMP)_2_(µ-O)Br_4_]^2+^ hexameric unit. Color scheme: (**a**) phosphorous, oxygen, nitrogen, carbon, and bromide are represented as gold sticks whereas copper as gold spheres; (**b**) green, gold, pink and brown colors are used to underline the four enlaced hexameric units; (**c**) copper, cyan spheres; phosphorous, orange; oxygen, red; nitrogen, blue; carbon, grey; bromide, brown. Hydrogen atoms and lattice water molecules have been omitted for the sake of clarity.

The elongated coordination with the exocyclic O(2) oxygen atoms leads to a distorted octahedral geometry around the terminal copper atoms, encompassing the bipy molecules and two coordinated bromide ions. The Cu–O [1.933(12)-2.741(12) Å], Cu–N [1.983(16)–2.036(8) Å] and Cu–Br [2.432(3)–2.822(3) Å] distances fall in the range previously reported for similar compounds [52,74]. 

Pyrimidine aromatic rings of the CMP ligand are planar as expected and exhibit an *anti*-configuration with respect to the ribose. The sugar unit for the two crystallographically independent cytidine monophosphate shows bent C(3’)-*eso* conformation. The conformation of the C(4’)-C(5’) bond is *gauche-gauche*. The chiral centers C(1’), C(2’), C(3’), and C(4’) have *R, R, S, R* configuration.

As stated above, the supramolecular assembly of the hexameric units is evidently modulated by π-π stacking, where pyridine rings from the bipy modulator, demonstrate their strength and action. These interactions are not only intramolecular, supporting the tetranuclear butterfly-shaped core (centroid-plane distances of 3.39, 3.43, 3.41, and 3.47 Å with deviation angles between involved planes of 6.7, 1.3, 4.5, and 3.5°, respectively) (Figure 2) but are at the origin of the porous network, generated by intermolecular interactions amid terminal bipy ligands of the hexameric units with a butterfly-shaped copper core of adjacent units (centroid-plane distances of 3.36, 3.38, 3.28, and 3.49 Å and angles between overlying planes of 5.1°, 3.4°, 11.3°, and 1.1°) (Figure 2). Indeed, it is the geometrical match between the aromatic ligands of the tails and tetranuclear cores, ensuring the π···π stacking supramolecular contacts, which makes CMP bent to generate alternating hydrophobic (A in Figure 3) and hydrophilic chiral nanopores (B and C in Figure 3), with an aperture of 1.2 and 0.8 nanometers, respectively, filled by crystallization water molecules (Figure 4). The hydrophobic pores contain both acetal oxygen atoms from the ribose moiety and hydrophobic pyridine ligands, while the hydrophilic pores are decorated by the hydroxyl groups of the sugar moiety pointing towards the pores (Figure 3 and Figure 4).

**Figure 2 molecules-26-04594-f002:**
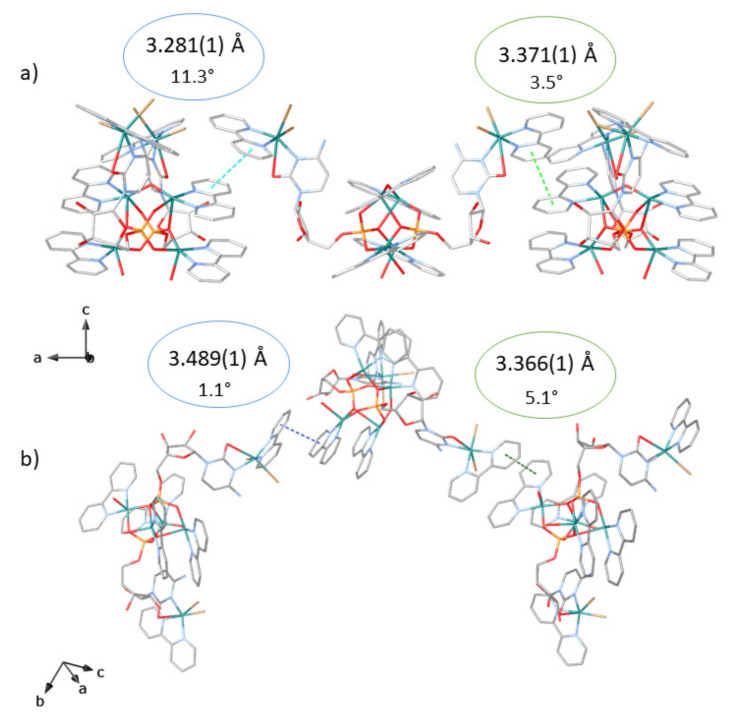
Perspective view of π-π stacking interactions within neighboring hexanuclear fragments: (**a**) (interplanar distances and dihedral angles of 3.281(1) Å (light blue dashed lines) and 11.3°, and 3.371(1) Å (light green dashed lines) and 3.5°, respectively); (**b**) (interplanar distance and dihedral angles of 3.489(1) Å (light blue dashed lines) and 1.1°, and 3.366(1) Å (light green dashed lines) and 5.1°, respectively). Copper, cyan spheres; phosphorous, orange; oxygen, red; nitrogen, blue; carbon, grey; bromide, brown. Hydrogen atoms and lattice water molecules have been omitted for the sake of clarity.

**Figure 3 molecules-26-04594-f003:**
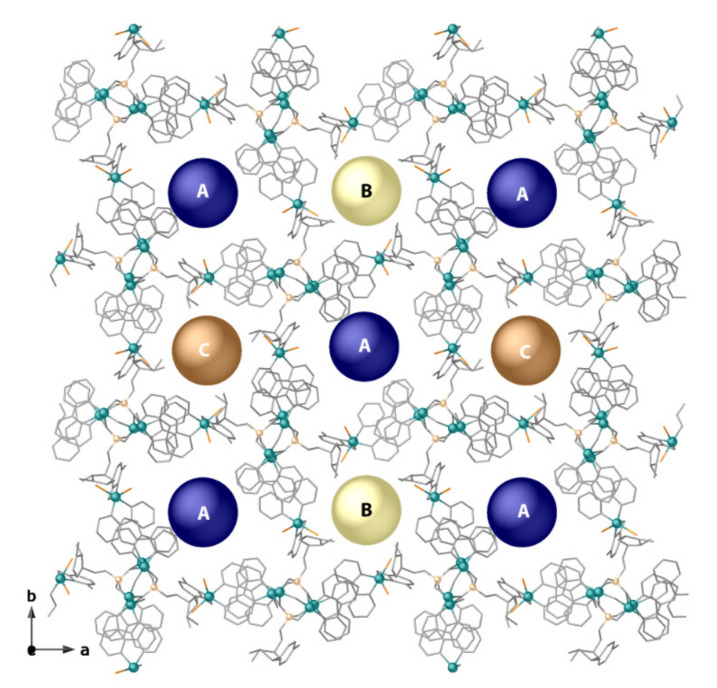
View along *c* crystallographic axis of the hydrophobic (**A**) and hydrophilic chiral nanopores (**B**,**C**), highlighted by dummy atoms (blue, gold, and brown for different types of pores). Color scheme: copper, cyan spheres; phosphorous, orange spheres; oxygen, nitrogen, carbon, grey sticks; bromide, brown sticks. Hydrogen atoms and lattice water molecules have been omitted for the sake of clarity.

**Figure 4 molecules-26-04594-f004:**
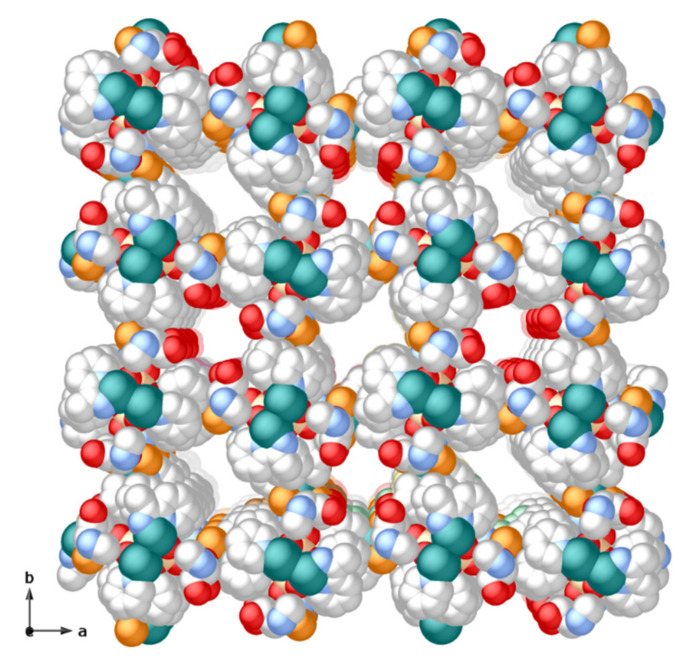
Perspective view along the *c* crystallographic axis of a fragment of **CMP MOF 1** porous structure as a space-filling model (atoms are represented with van der Walls radii). Alternating hydrophobic and hydrophilic nanopores exhibit virtual diameters of 1.2 and 0.8 nanometers, respectively. Color scheme: copper, cyan; phosphorous, orange; oxygen, red; nitrogen, blue; carbon, grey; bromide, brown. Hydrogen atoms and lattice water molecules have been omitted for the sake of clarity.

This overall crystal packing featuring porous channels decorated by the ribose hydroxyl groups provides the nanostructure with the potential for chiral molecular recognition. Indeed, the chirality of the channels in these kinds of compounds derived from chiral pure nucleotide, is ensured by the chiral sugar moieties, which pointing within the pores, might impose supramolecular chirality as well, simply due to weak interactions, such as H-bonds, involving hydrophilic hydroxyl groups and water molecules filling the channels. 

The nature and size of pores allow for the hosting of a large number of solvent water molecules, stabilized by an intricate network of hydrogen bonds. Indeed, the resulting potential solvent accessible area with a volume of 3659.3 Å^3^ accounts for 31.5% of the unit cell volume (11615.6 Å^3^).

Although not all water molecules were detected by thermogravimetric analysis (TGA) (Figure 5) due to the huge pores and diffuse electron density, some of them have been modeled. It is evident that in the hydrophilic pores the hydroxo groups from ribose of CMP are pretty involved in anchoring water clusters to the walls (O···O distances varying in the range 2.40(3)–3.39(4) Å). Further weak van der Waals intra- and intermolecular interactions support the overall self-assembly by means of Br···H–N (Br···N distances varying in the range 3.29(1)–3.34(1) Å, see Appendix A) and Br···π interactions involving bromide ions and bipy modulator with averaged distance of 4.4(1) Å (Appendix A). 

It is worth underlining that, from a structural point of view, the final porous framework of the **CMP MOF 1** is generated by π-π stacking interactions established between the bipy (Figure 1a,b). However, all the potential H-bonds strongly support the structure from a chemical point of view, contributing to stabilize the overall network.

Diverse factors are at the origin of the geometrical match achieved and are essential for displaying the potential of these kinds of forces. Even if much remains to be done for their understanding, it is clear that a synergistic effect of structural features of building units and their potential for multiple interactions pathways is able to modulate, sometimes in an exquisite manner, their final supramolecular assemblies. In **CMP MOF 1**, the tetranuclear core, well stabilized by both intramolecular π-π and hydrogen bonding interactions, is rigid and robust enough to feed the further interaction with the more flexible tails of the hexanuclear core. 

The crystal structure of **CMP MOF 1** was deconstructed by applying the conception of the simplified underlying net. The TOPOSPRO software [75,76] has been used in order to get a topological analysis. The butterfly-shaped copper cores have been represented as a node connected to adjacent nodes at a distance of ca 18 Å. Thus, the overall structure unfolds to a very unusual **8T6** network. Indeed, this topology has occurred only one time in the ToposPro [75] suggesting the potential of π-π stacking as supramolecular interaction to construct unexpected and/or new topologies.

**Figure 5 molecules-26-04594-f005:**
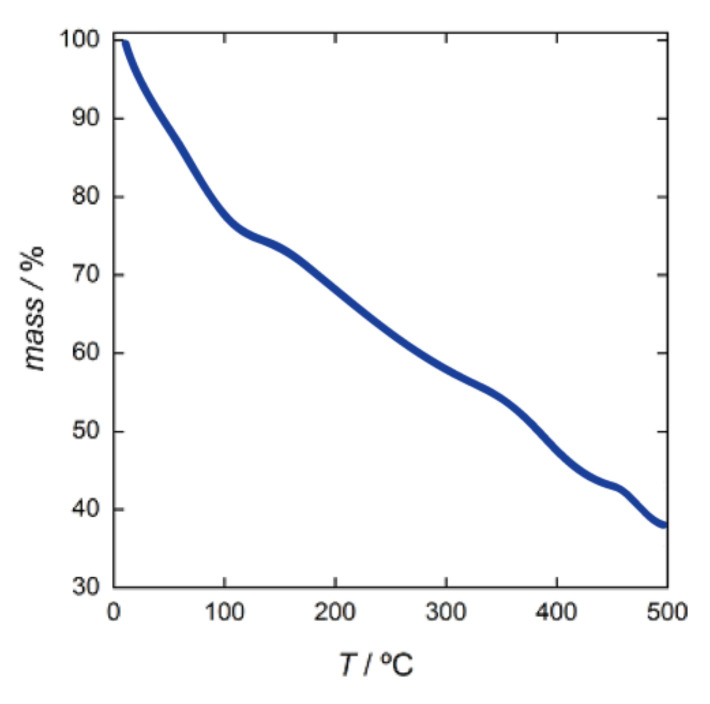
TGA of **CMP MOF 1** under dry N_2_ atmosphere.

## 3. Materials and Methods

All reagents were purchased from commercial sources and used as received.

Elemental analyses (C, H, and N) were performed at the microanalysis service of the Dipartimento di Chimica e Tecnologie Chimiche of the Università della Calabria (Italy). FTIR spectra were recorded on a Nicolet-6700 spectrophotometer as KBr pellets. The thermogravimetric analysis was performed on crystalline samples under a dry N_2_ atmosphere with Perkin-Elmer equipment with a thermobalance operating at a heating rate of 10 °C min^−1^. 

### 3.1. Synthesis of Compound ***CMP MOF 1***


An aqueous solution of CuBr_2_ (66,9 mg, 3 mmol, solvent volume 3 mL) was added dropwise and under gentle stirring to an aqueous solution of Na_2_CMP (32,3 mg, 1 mmol, solvent volume 2 mL). Very quickly, a light blue powder precipitated, after that an ethanolic solution of 2,2’-bipy (3 mmol, 46,8 mg, solvent volume 3 mL) was added dropwise. The resulting mixture was stirred for 30 minutes until the complete solubilization of the blue powder. Well-formed green crystals of CMP MOF 1, suitable for X-ray diffraction, were formed by very slow evaporation, then the solution was left tostand at air and room temperature for a week. Yield 84%; anal. calcd. for C78H164Br6Cu6N18O63P2 (3284.90): C, 28.52; H, 5.03; N, 7.68. Found: C, 27.01; H, 5.06; N, 7.92%. IR (KBr): IR (KBr): *ν* = 1678vs, 1646vs and 1352s cm^−1^ (C=O) from CMP and 1471s and 1445 cm^–1^ (C=C) from bipy.

### 3.2. Thermogravimetric Analysis 

The water content of **CMP MOF 1** was determined by thermogravimetric analysis (TGA) under a dry N_2_ atmosphere. It shows a fast mass loss from room temperature to ca. 310 ℃, evidencing a short plateau in the mass loss (in the range 100 to 140 ℃) until decomposition starts. The estimated percentage weight loss value of 25% (Figure 5) corresponds to ca 46 H_2_O molecules per formula. The fast-decreasing profile observed by TGA is due to the fast loss of solvent, even at the origin of structure collapse upon exposure to air for more than 1 hour, which prevents any possibility for the experimental study of permanent porosity. It is one of the main limitations of supramolecular bioMOFs due to their poor mechanical robustness, which results in the collapse of the structure upon solvent removal. However, if this is of course an issue for gas storage, where the porous materials must be activated, it is not for materials potentially useful for chiral recognition as for **CMP MOF 1**. It is worth underlining that chiral recognition is a dynamic process, normally conducted in solvents and evacuation is not imperative since the solvent molecules that reside in the pores can be gradually exchanged with the chiral guests in solution. 

### 3.3. X-ray Powder Diffraction 

A polycrystalline sample of **CMP MOF 1** was deposited on a flat plate with a 5 cm diameter prior to being mounted on a Bruker D2 PHASER Diffraction System with Cu-Kα radiation (λ = 1.54056 Å). Five repeated measurements were collected at room temperature (2θ = 2–50°) and merged in a single diffractogram. The experimental powder X-ray diffraction (PXRD) patterns of a polycrystalline sample of **CMP MOF 1** confirm the purity and homogeneity of the bulk sample (Figure 6) and that the 3D anionic network does not experiment significant phase transitions in the range 100 to 298 K. 

### 3.4. Single Crystal X-ray Diffraction 

A crystal of **CMP MOF 1** was selected and mounted on a MITIGEN holder in Paratone oil and very quickly placed on a liquid nitrogen stream cooled at 100 K to avoid the possible degradation upon dehydration. Diffraction data were collected on a Bruker-Nonius X8APEXII CCD area detector diffractometer using graphite-monochromated Mo-Kα radiation (λ = 0.71073 Å). The data were processed through the SAINT reduction and SADABS [77] multi-scan absorption software. The structure was solved with the SHELXS structure solution program, using the Patterson method. The model was refined with version 2018/3 of SHELXL against F^2^ on all data by full-matrix least squares [78,79]. 

All non-hydrogen atoms were refined anisotropically, except for C1B’, C2B’, and C28. The solvent molecules were disordered and some of them refined with 0.5 (O13W and 015W) or 0.25 (O19W and O21W) of occupancy factors and they have been only in part modeled, the quite large channels featured by this MOF likely account for that. In fact, only water molecules pseudo-coordinated to copper or well enlaced by means of strong and medium H-bonds have been modeled. Any attempts to locate and model the highly disordered guest molecules in the pores were unsuccessful.

For that reason, in **CMP MOF 1**, the contribution to the diffraction pattern from the highly disordered water molecules located in the voids was subtracted from the observed data through the SQUEEZE method, implemented in PLATON [80]. The total potential accessible voids calculated by PLATON has a volume of 3659.3 Å^3^ and accounts for 31.5% of the unit cell volume (11615.6 Å^3^). In the voids per unit cell, SQUEEZE estimated a total count of 1859 electrons, which is in agreement to be 46 water molecules (Z = 4; H_2_O = 10e−; 1859e−/4 = 464.7e- ≈ 46H_2_O).

The hydrogen atoms of the ligand were set in calculated positions and refined as riding atoms whereas for water molecules, they were neither found nor calculated.

Crystal data for C78H164Br6Cu6N18O63P2 (*M* = 3284.90 g/mol): tetragonal, space group *P*4 (no. 75), *a* = 31.9472(12) Å, *c* = 11.3809(4) Å, *V* = 11615.6(10) Å^3^, *Z* = 4, *T* = 100(2) K, *μ*(MoK_α_) = 0. 0.71073 mm^−1^, *ρ*_calc_ = 1.878 g/cm^3^, 166,532 reflections measured (0.901° ≤ *θ* ≤ 26.438°), 23,831 were unique (*R*_int_ = 0.0644, R_sigma_ = 0.0721) which were used in all calculations. The final *R* was 0.0891 (I > 2*σ*(*I*)) and *wR* was 0.2417 (all data). A summary of the crystallographic data and structure refinement for crystal structure is also given in Appendix A. The comments for the alerts A and B are described in the CIF using the validation response form (vrf). The CCDC reference number is 2092761. 

The final geometrical calculations on free voids and the graphical manipulations were carried out with PLATON [80] implemented in WinGX [80,81], and CRYSTAL MAKER programs [81], respectively.

## 4. Conclusions

We report the rational design of a new chiral supramolecular BioMOF. It represents an uncommon example of MOF constructed exploiting the use of a ligand as a modulator to entail π···π supramolecular interactions. Canonical hydrogen bonds, in **CMP MOF 1**, represent only auxiliary interactions to boost its final nanoporous architecture. Mutually, intra- and intermolecular π···π interactions are implicated in the construction of the **CMP MOF 1** porous supramolecular framework. The present results represent rare examples of a judicious exploitation of these interactions to modulate the conformational preferences of a molecule, depending on its chemical nature. Although π···π interactions are over-documented forces, a priori applications in rational synthesis for new porous frameworks with new topologies are relatively bare, even if it might offer a powerful tool, especially when synergistically coupled with intramolecular and intermolecular hydrogen-bonding interactions. Indeed, the examples previously reported have been underlined in post facto analyses of crystallographic information, whereas there are a modest number of examples that have been reported where potential π-system donors have been exploited in a more judicious manner. 

## Data Availability

Crystallographic details in CIF format are available at https://www.ccdc.cam.ac.uk/ website.

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
