# Peer review of "A Nanoporous Supramolecular Metal–Organic Framework Based on a Nucleotide: Interplay of the π···π Interactions Directing Assembly and Geometric Matching of Aromatic Tails"

_molecules, 2021, doi:10.3390/molecules26154594_

Round 1
Reviewer 1 Report
This is a nice paper reporting a nanoporous supramolecular metal-organic framework. The authors used the self-assembly of cytidine 5´-monophosphate (CMP) nucleotide and copper metal ions, with π-π stacking as the leading factor, and hydrogen bonding as the assistance, to construct CMP MOF 1. This unique MOF structure is a rare example of using supramolecular interactions to modulate the conformational preferences of a molecule. I think this is an interesting work, and it provides some new insights into the construction of porous frameworks. I recommend its publication in Molecules after following minor issues be appropriately addressed.
- Page 2, line 44-59. The subject of this work is MOF, but this paragraph talks too much about HOFs.
- Page 2, line 56. The author states that there are many literature reports that maintain the supramolecular framework mainly by π–π interactions. These two important papers should be cited: Science Advances, 2020, 6, eaax9976 and Angew. Chem. Int. Ed. 2021, 60, 7148.
- Page 6, line 185 and page 6, line 200, please unify the description of π-π stacking.
- For figure 4 and figure 5, for better reading and understanding, the author should mark the distances and angles in the figures.
- Page 6, line 206. TGA is not in Figure 6, please be careful. Meanwhile, please make it clear the temperature corresponds to 25% weight loss is 310 K or 310 °C.
Reviewer 2 Report
The manuscript “A nanoporous supramolecular metal-organic framework based on a nucleotide: Interplay of the π···π interactions directing assembly and geometric matching of aromatic tails.” By Donatella Armentano and collaborators presents the crystallographic analysis of a newly synthesized copper(II) complex bearing as ligands cytidine 5’-monophosphate and bpy. The resulting structure forms, in the crystalline state, a supramolecular arrangement including channels filled with crystallization water. Surrounding them, the complex forms a network held together by the coordination interactions together with supramolecular contacts. The authors assign π-stacking interactions as the main forces driving the formed structure. However, the arguments supporting the claim are not sufficient and the manuscript has to be revised, for instance:
- The main claim of the authors is that the supramolecular crystalline arrangement is driven by the π-π stacking interactions between the bpy moieties, however, the actual structure contains a lot of water and presents an intricated network of hydrogen bonds. The authors say in line 215 that all the H-bonds are auxiliary forces to pi stacking interactions, but in general this is the opposite and the regular prevalence of hydrogen bonds over π-stacking is well documented (see ref). The authors should justify this claim. Moreover, the authors state that the structure readily collapses upon dehydration, which supports an important role of the H-bond patterns in supporting the structure.
CrystEngComm 2009, 11 (6), 1122. https://doi.org/10.1039/b820791g
Inorg. Chem. 2020, 59 (13), 8667–8677. https://doi.org/10.1021/acs.inorgchem.9b03131
RSC Adv. 2014, 4 (48), 25018–25027. https://doi.org/10.1039/C4RA04028G
- The discussion about the claimed chirality of the channels is superficial, the authors should elaborate on it. If the structure presents some chirality, can it be observed by circular dichroism?
- The analysis elemental of the compound suggests it is not pure. Although the H and N % are in agreement with the proposed formula, the carbon is by far out of range.
- The TGA is weird and doesn’t correspond to the explanation given, for instance at 310 K (37°C) the mass is in a clear descendent slope, and there is not such a plateau. There’s a short plateau beyond 100°C, which makes sense with water loss. But the mass continues going down in an almost constant manner.
- Additional characterization should be provided in order to show the purity and prove the identity of the compound. In particular, mass spectra, and EPR could be helpful.
Overall, although the crystalline structure found is interesting the analysis of it presented by the authors is flawed and biased, there is no evidence supporting the main contribution of the π-stacking interactions to the assembled structure, the clear incidence, and importance of hydrogen bonding is refuted a priori, what is unacceptable. The authors should provide experimental evidence supporting their claims. Additionally, some other points have to be taken into account:
Minor Points
- The figures do not state the omission of water molecules in the structures.
- Figures 1, 2, and 3 are redundant, figure 1 includes simplified versions of figures 2 and 3. I suggest keeping just figure 1 showing the perspectives as in 2 and 3. Similar for figure 4 and 5 and 6, 7 and 8, yes, they are nice but they are not really needed, it would be better to have just one good figure with the needed labels and annotations which properly illustrates what is wanted in detail than three separated figures that show the same in a different way.
- The color codes are hard to follow and present errors in several of the figures, for example in Line 108, oxygen should not be in the color list. I suggest keeping as pairs the color code, i.e. red, oxygen; blue, nitrogen; black, carbon; etc.
- Scheme 1 image quality is too low, increase the resolution and text size on the structures and increase the size of the crystal structure.
- Details:
Abstract line 21 should say: These kinds of materials… candidates
Line 86 bipyridine
References
I consider that the work of Prof. M. W. Hosseini is fundamental in the study of self-assembly in the crystalline phase, and it is not cited.
About microporous chiral materials, I think the following review has to be cited:
Materials Today, 19, 9, 2016, 503-515 https://doi.org/10.1016/j.mattod.2016.03.003
In line 71 the authors talk about the nucleotide-metal complexes being problematic to prepare and present few examples in table S1. Actually, the topic is much more abundant in the practice and the authors should refer and address more comprehensive literature, for example:
Coord Chem Rev, 432, 2021, 213705 https://doi.org/10.1016/j.ccr.2020.213705
Coord Chem Rev, 292, 2015, 107-143 https://doi.org/10.1016/j.ccr.2015.02.007
Chin.Sci.Bull. 47, 1–9 (2002). https://doi.org/10.1360/02tb9001
Chem. Soc. Rev., 1993,22, 255-267 https://doi.org/10.1039/CS9932200255
Round 2
Reviewer 2 Report
The revised version of the manuscript “A nanoporous supramolecular metal-organic framework based on a nucleotide: Interplay of the π···π interactions directing assembly and geometric matching of aromatic tails.” Shows important improvements and most of the points raised in the previous revision have been reasonably addressed. I suggest its acceptance although I consider a few more improvements can be made prior to the final publication:
- The quality of the graphics is still awful, I encourage the authors to try to get higher resolution images, for that diverse rendering options are normally included in most of the visualization packages that would allow high-resolution images. It is a pity that some of these beautiful images don't have the best quality. Regarding the structure drawing also in ChemDraw (if it is being used), I recommend using the “save as” options menu to save as an image (tiff maybe) and select in the “options” menu the desired resolution (I would recommend at least 500 ppm).
- Although the introduction is quite concise I think it lacks some contextualization in the use of copper(I) or copper(II) in the construction of self-assembled materials, some literature examples include:
- Inorg. Chem. 2008, 47, 1, 162–175 https://doi.org/10.1021/ic701669x
- Dalton Trans., 2010,39, 8646-8651 https://doi.org/10.1039/C000180E
- Bulletin of the Chemical Society of Japan, 2003, 76:4, 789-797 https://doi.org/10.1246/bcsj.76.789
- Chem. Eur. J. 2021, 27, 8308. https://doi.org/10.1002/chem.202100865
